# In Vitro Models for Studying Chronic Drug-Induced Liver Injury

**DOI:** 10.3390/ijms231911428

**Published:** 2022-09-28

**Authors:** M. Teresa Donato, Gloria Gallego-Ferrer, Laia Tolosa

**Affiliations:** 1Unidad de Hepatología Experimental, Instituto de Investigación Sanitaria La Fe, 46026 Valencia, Spain; 2Departamento de Bioquímica y Biología Molecular, Facultad de Medicina y Odontología, Universidad de Valencia, 46010 Valencia, Spain; 3Centro de Investigación Biomédica en Red de Enfermedades Hepáticas y Digestivas (CIBERehd), Instituto de Salud Carlos III, 28029 Madrid, Spain; 4Center for Biomaterials and Tissue Engineering (CBIT), Universitat Politècnica de València, 46022 Valencia, Spain; 5Biomedical Research Networking Center on Bioengineering, Biomaterials and Nanomedicine (CIBER-BBN), 46022 Valencia, Spain

**Keywords:** chronic hepatotoxicity, cellular models, mechanisms

## Abstract

Drug-induced liver injury (DILI) is a major clinical problem in terms of patient morbidity and mortality, cost to healthcare systems and failure of the development of new drugs. The need for consistent safety strategies capable of identifying a potential toxicity risk early in the drug discovery pipeline is key. Human DILI is poorly predicted in animals, probably due to the well-known interspecies differences in drug metabolism, pharmacokinetics, and toxicity targets. For this reason, distinct cellular models from primary human hepatocytes or hepatoma cell lines cultured as 2D monolayers to emerging 3D culture systems or the use of multi-cellular systems have been proposed for hepatotoxicity studies. In order to mimic long-term hepatotoxicity in vitro, cell models, which maintain hepatic phenotype for a suitably long period, should be used. On the other hand, repeated-dose administration is a more relevant scenario for therapeutics, providing information not only about toxicity, but also about cumulative effects and/or delayed responses. In this review, we evaluate the existing cell models for DILI prediction focusing on chronic hepatotoxicity, highlighting how better characterization and mechanistic studies could lead to advance DILI prediction.

## 1. Introduction

Drug-induced liver injury (DILI) is a multifactorial potentially life-threatening adverse reaction to prescribed drugs and other chemicals [1]. DILI is both a cause of drug attrition in developmental phases and a major cause of drug withdrawal after commercialization, which poses a challenge for the pharmaceutical industry. Approximately, 25% of drug withdrawals [2] and 20% of black box warnings [3] of prescription medications are due to DILI. The main reason explaining this susceptibility is probably its anatomical proximity to the gastrointestinal tract and the portal vein supply [4] as well as the liver’s central role in the metabolism of xenobiotics. Thus, it would be desirable to remove drugs with an elevated risk of causing DILI early in the drug development process. Screening large sets of compounds in vivo for efficacy and toxicity is too costly and time-consuming [5]. Additionally, animal studies face important limitations since there are substantial differences between experimental animals and humans in drug metabolism and pharmacokinetics [6,7]. To overcome these limitations, human cell models emerged as an alternative for the early detection of hepatotoxicity before compounds enter clinical trials. For the evaluation of acute hepatotoxicity, a wide range of cell models has been proposed [8,9]; however, animal models are still mainly used for the assessment of long-term effects. Although in vitro models have many limitations, current legislation strongly encourages the use of new alternative methodologies (NAMs) to predict chronic DILI because it implies reduced time and costs as well as to replace, reduce and refine animal use (3Rs) in safety and efficacy evaluations [10,11].

DILI reactions are commonly multifaceted events that require various interactions between hepatocytes and non-parenchymal cells (NPCs), such as hepatic stellate cells (HSCs) or Kupffer cells (KCs). Moreover, drugs can induce DILI through a variety of mechanisms [12], including mitochondrial injury, oxidative stress induction, steatosis, cholestasis and immune-mediated reactions. Primary human hepatocytes (PHHs) are the gold standard cellular model for studying DILI due to their physiological relevance; however, when cultured in conventional monolayer two-dimensional (2D) cultures, they rapidly lose the majority of their functions [9,13,14]. As a consequence, the use of conventional 2D mono-cultures for DILI detection is limited because of the complexity and variety of toxicity mechanisms and the prolonged periods required to mimic the standard drug treatment regimes in humans [15]. In recent years, researchers have focused on the development and optimization of new cell culture systems that better reflect the in vivo physiology. These new models include sandwich-cultured PHHs, co-cultures and three-dimensional systems such as spheroids or microfluidic devices, and often allow the maintenance of the PHH phenotype for longer periods, enabling the testing of chronic hepatotoxicity, and providing information not only about toxicity, but also about cumulative effects and/or delayed responses.

Susceptibility to DILI is probably a consequence of multiple factors related to the genetic background, underlying diseases and associated medications of the patient as well as the pharmacology of the drug [16]. Additionally, it should be considered that a minor percentage of DILI cases can become chronic, with various risk factors such as age and metabolic risk factors [17]. Thus, new cell models that better reflect variability are also highly desirable.

Although there has been a number of comprehensive reviews on different cell models for hepatotoxicity studies, this review focuses on the evaluation of chronic hepatotoxicity and the existing in vitro models to study it. To this purpose, we describe the major mechanisms implicated in DILI, highlighting the importance of chronic effects such as steatosis or cholestasis. We also exemplify some drugs that are known to produce chronic hepatotoxicity. Then, we discuss in depth the emerging in vitro hepatic cell culture platforms and the technological methods for evaluating chronic DILI.

## 2. Chronic Drug-Induced Liver Injury

A high heterogeneity is observed in clinical manifestations, severity and outcomes of DILI. Toxic liver damage shows has a wide spectrum of clinical presentations, ranging from asymptomatic and transient elevations in the serum levels of liver enzymes to fulminant liver failure. In clinical practice, consensus criteria based on alanine aminotransferase and alkaline phosphatase activity levels, and their relative ratio, are used to classify hepatotoxicity pattern as hepatocellular, cholestatic and mixed [18,19]. Hepatotoxicity is usually self-limited and a complete recovery of the patient, with no apparent sequels and normalisation of liver tests, is reached after the withdrawal of the culprit drug. However, severe and fatal cases of DILI can occur [19]. In some patients, the episode of liver injury does not resolve after the drug therapy is stopped and it can evolve to acute liver failure leading to death/liver transplantation or it can persist over time and become chronic [20,21].

There are no data on the global frequency of chronic hepatotoxicity, due to the limited number of studies performed to determine the long-term outcome of patients with DILI. Diverse national prospective studies reported a great variability in the geographical incidence of chronicity in patients with a diagnostic of DILI (ranging from 5.7 to 18.8% of DILI patients) [22,23,24,25,26]. These differences can be attributed, at least in part, to the lack of general consensus criteria to define chronic DILI. The concept of chronic DILI has undergone a continuous evolution over the last decades. The first definition included cases of persistent abnormalities in liver tests for more than 3 months after DILI onset [27]. Later on, alterations beyond 6 months were considered for cholestatic/mixed injury [22] and, more recently, 1 year was proposed as the best cut-off for chronic DILI [17,18].

The list of agents able to induce chronic DILI includes drugs from many therapeutic groups (e.g., antibiotics, anticonvulsants, antidepressants, antineoplastic agents, antihypertensives, immunomodulatory agents, NSAIDs and cardiovascular drugs) as well as herbal and dietary supplements [19,22,24,28,29]. Drugs associated with chronic DILI are provided in Table 1.

**Table 1 ijms-23-11428-t001:** Drugs inducing major types of chronic hepatocellular or cholestatic liver injury.

Autoimmune-like Drug-Induced Liver Injury [20,30,31,32]	Hepatic Steatosis and/or NASH [30,33,34,35,36]	Phospholipidosis/Hepatocellular Deposits Cytoplasmic Inclusions [35,36,37,38,39]	Cholestasis or Vanishing Bile Duct Syndrome [22,30,32,36,40,41,42]
α-methyldopa Allopurinol Benzarone Chlorpromazine Dantrolene Diclofenac Dihihydralazine Fluoxetine Halothane Indomethacin Interferon β Isoniazid Minocycline Nitrofurantoin Phenytoin Statins: Atorvastatin, Rosuvastatin, Simvastatin Sulfonamide Terbinafine Tienilic acid TNF-α antagonists: Adalimumab, Infliximab, Etanercept	5-fluorouracil Amineptine Acetylsalycilic Amiodarone Apo-B inhibitors: Mipomersen, Limitapide Doxycycline Fialuridine Glucocorticoids (Dexamethasone) Ibuprofen Irinotecan Methotrexate Nifedipine NRTIs: Zidovudine, Didanosine, Stavudine Perhexiline Tamoxifen Tetracycline Tianeptine Valproic acid	Amiodarone Amikacin Amitriptyline Chlorpromarizine Citalopram Clomipramine Clozapine Desipramine Fluoxetine Haloperidol Hydroxyzine Imipramine Ketoconazole Maprotiline Perhexiline Sertraline Tamoxifen Thioridazine Tilorone Zimelidine	Anabolic-androgenic steroids Amitriptyline Amoxicillin–clavulanic acid Azathioprine Azithromycin Bosentan Captopril Carbamazepine Chlorpromazine Cyclosporine Erythromycin Ethynyl estradiol (oral contraceptives) Flucloxacillin Griseofulvin Haloperidol Itraconazole Levofloxacin Nimesulide Phenytoin Terbinafine

NRTIs: nucleoside reverse transcriptase inhibitors.

The hepatic lesions induced by drugs are heterogeneous and may affect any liver cell type (hepatocytes, cholangiocytes and endothelial cells). Although persistent DILI may present different profiles of liver injury, it has been reported that cholestatic and mixed damage require a longer time to normalize and patients with a cholestatic pattern have a higher risk for developing chronic DILI [22,24]. Aged patients, severity of liver damage at onset, prolonged duration of drug therapy prior to the diagnosis of DILI or drug re-exposure are additional factors that may contribute to chronicity [17,24,28].

Chronic DILI can be classified based on overall pathologic manifestations and histological findings into several clinical phenotypes that often share features with many other liver diseases induced by nontoxic causes. The list of phenotypic subtypes of chronic DILI includes autoimmune-like DILI, steatosis, steatohepatitis, phospholipidosis, bland cholestasis, vanishing bile duct syndrome, sinusoidal obstruction syndrome, nodular regenerative hyperplasia, peliosis hepatitis, hepatoportal sclerosis, granulomatous hepatitis and hepatic neoplasms [30,43]. This classification may partially overlap with the pattern of liver injury (hepatocellular and cholestasis) and includes other types of liver injury, such as drug-induced vascular liver lesions and obstruction disorders [21,29,43,44,45]. In this section, special attention is paid to four major clinic phenotypes representative of chronic DILI and information on their pathological features. The mechanisms involved are exemplified in Figure 1.

### 2.1. Autoimmune-like DILI

Many drugs have been related to cases of liver damage with autoimmune features, with nitrofurantoin and minocycline being the most commonly involved drugs [40,46]. Patients with autoimmune-like DILI present similar clinical, biochemical and histological characteristics in comparison with patients with idiopathic autoimmune hepatitis [40]. Severe portal inflammation, lymphoplasmacytic infiltrates, rosette formation and severe focal necrosis are major histological characteristics of this type of liver injury [29]. Antinuclear antibodies, smooth muscle antibodies and high IgG levels are found in autoimmune-like DILI subjects at a similar proportion than in autoimmune hepatitis patients [40]. A major difference among both groups of patients is that autoimmune-like DILI shows a general favourable outcome after the discontinuation of the implicated drug and corticosteroid therapy could be discontinued without relapse [46].

The mechanism responsible of this type of DILI remains largely unknown. The potential role of reactive metabolites generated during the hepatic metabolism of the drug has been proposed. Electrophilic metabolites generated by cytochrome P450 (CYP) enzymes can covalently bind to hepatocyte proteins and these drug–protein adducts can be recognized as neoantigens by the immune system. This hypothesis was supported by the identification of circulating autoantibodies directed against hepatic microsomal CYP proteins in patients with autoimmune hepatitis induced by tienilic acid, dihydralazine or halothane [47,48,49]. However, a similar detection of specific autoantibodies has not been demonstrated for other drugs known to induce autoimmune-like DILI. Bioactivation to reactive metabolites has been reported for many drugs, most of which do not induce autoimmune-type DILI, so other risk factors should be considered [50]. The final DILI phenotype likely results from a combination of drug properties and patient-specific susceptibility [45,51]. Among the host-related factors, those that influence drug disposition and pharmacokinetics, inflammatory signalling pathways and the immune response are of particular relevance for autoimmune-like DILI. Various class I and II HLA genotypes, and polymorphic variants of drug-metabolizing enzymes (NAT2, UGTs and GST), drug transporters (BSEP, MRP2 and MRP4) or enzymes involved in antioxidant defence (SOD2 and GPX1) have been associated with an increased predisposition to idiosyncratic DILI (iDILI) [16,45,51,52,53,54,55,56]; however, no genetic factor has been definitively associated with the risk of autoimmune-like DILI.

Long-term exposure to nitrofurantoin, an antimicrobial agent widely used for the treatment of recurrent urinary tract infections, has been associated with a broad spectrum of hepatotoxic effects that could result into chronic and severe liver damage and even death [57]. Although the mechanism underlying chronic hepatotoxicity induced by nitrofurantoin remains unclear, it is likely an immune-related process. This is supported by the higher predisposition in females, the presence of auto-antibodies and histological evidences [57,58]. Aged patients, impaired renal function and long-term exposure to the drug are also considered risk factors. Diverse studies performed to clarify the mechanism of nitrofurantoin hepatotoxicity suggested the potential role of drug metabolism [59,60,61]. Structure toxicity relationship studies revealed the key role of the nitrofuran moiety of nitrofurantoin in its bioactivation to toxic species [59,60]. In particular, an in vitro study using liver microsomes and primary hepatocytes evidenced that nitro reductive metabolism of the drug leads to the formation of nitroso and hydroxylamine intermediates able to deplete GSH levels and induce cytotoxicity [60]. The study revealed that the higher electrophilic reactivity of nitrofurantoin and other nitrofuran-containing species compared to other nitroaromatic compounds may favour the hepatotoxicity potential of this drug.

### 2.2. Drug-Induced Steatosis

Steatosis, defined as the excessive accumulation of fat in the liver, is a frequent histopathological finding in liver biopsies from DILI patients [21]. Lipids, mainly triglycerides, may accumulate in the cytoplasm of hepatocytes as small or large fat droplets leading to micro- or macrovesicular steatosis, respectively. Drug induced-microvesicular steatosis is a rare and potentially severe liver lesion that can result in acute liver failure and has often been related to mitochondrial injury [62,63]. This type of liver damage has been reported after exposure to tetracycline, amiodarone, valproic acid or antiviral drugs (fialuridine and NRTIs) in a few patients who may likely have an underlying mitochondrial dysfunction [33,34,63]. Macrovesicular steatosis is more frequently associated with drugs, and, in most cases, it represents a benign condition in the short-term without signs of inflammation [21]. However, it can become chronic after prolonged exposure to drugs and progress into more severe lesions, such as steatohepatitis, fibrosis or cirrhosis. Amiodarone, perhexiline, irinotecan, methotrexate and tamoxifen are among drugs able to induce macrovesicular steatosis.

Several mechanisms have been implicated in drug-induced steatosis, with mitochondria dysfunction playing a key role. Drugs may reduce fatty acid oxidation in the liver by the direct inhibition of mitochondrial β-oxidation enzymes (amiodarone, tamoxifen, tianeptine, amineptine, tetracycline, doxycycline, methotrexate and perhexiline), impairment of the entry of fatty acids into the mitochondrial (valproic acid and ibuprofen), disruption of mitochondrial respiratory chain (amiodarone, tamoxifen and perhexiline zidovudine) or damage to mitochondrial DNA (NRTIs, fialuridine and tamoxifen) [35,63]. The inhibition of microsomal triglyceride transfer protein and altered export of lipids from the liver as very low-density lipoproteins (amiodarone, amineptine, tetracycline, stavudine and tianeptine) or increased lipid synthesis/uptake (tamoxifen, tetracycline and methotrexate) have been pointed as additional mechanisms [62,64].

Drug injury to mitochondria is also of relevance in the possible progression of steatosis towards more severe conditions [34]. Alterations in mitochondrial metabolism associated with the presence of the drug can contribute to reduce ATP synthesis, increase the generation of excessive reactive oxygen species and deregulate the antioxidant defence systems, which may contribute to exacerbating mitochondrial dysfunction. Thus, the depletion of intracellular ATP levels and oxidative stress can cause further hepatocyte injury. It is worth noting that hepatic steatosis is a highly prevalent liver disorder, particularly in Western countries, and is the result of multiple mechanisms and risk factors (obesity, metabolic syndrome, type 2 diabetes and medications). In this context, it has been reported that some drugs (e.g., tamoxifen, methotrexate and valproic acid) may exacerbate liver damage in patients with underlying mitochondrial disorders or pre-existing steatosis due to causes other than drugs [33,65,66].

### 2.3. Drug-Induced Phospholipidosis

Phospholipidosis is a lipid storage disorder resulting in the excessive intracellular accumulation of phospholipids in the affected tissues. This is a recurrent finding in toxicology studies and many drugs from different therapeutic groups (antidepressants, antianginal, antiarrhythmic, antimalarial, antibiotics or inhibitors of cholesterol synthesis) are able to induce phospholipidosis in the liver, lung, kidney and other tissues after long-term exposure [67]. This condition is usually associated with the so-called cationic amphiphilic drugs (CAD), a group of drugs that contain a hydrophobic ring and a hydrophilic amine group that can be protonated at acidic pH. Over 50 CAD, including highly prescribed agents such as amiodarone, chlorpromazine, chloroquine, desipramine or imipramine, have been reported as phospholipidosis inducers (Table 1).

The presence of concentric membrane-bound cytosolic inclusions from a predominant lysosomal origin, known as lamellar bodies, is a characteristic histological hallmark in drug-induced phospholipidosis. These lysosome-derived structures resemble intracytoplasmic deposits of phospholipids observed in Nieman–Pick disease, an inherited lysosomal metabolic disorder. Lamellar inclusion bodies are heterogeneous in terms of their origin, size and composition [68]. The pattern of accumulated phospholipids may differ among drugs and, in association with phospholipid and other lipidic molecules, the accumulation of the inducing drug may also be observed [69,70].

The mechanisms involved in drug-induced phospholipidosis remain unclear. Drugs may cause the accumulation of phospholipids either by reducing their degradation or increasing their biosynthesis. The direct inhibition of lysosomal phospholipase activity and/or the formation of drug-phospholipid complexes that are resistant to enzymatic degradation have been proposed to explain how CADs (e.g., amiodarone, chlorpromazine, chloroquine and desipramine) can affect phospholipid degradation [62]. CADs are amphiphilic molecules that in their protonated form can easily cross biological membranes. Once inside the lysosomal acidic compartment, CADs are deprotonated, acquire a positive charge and remain trapped. CADs can accumulate in lysosomes over time and electrostatically interact with negatively charged membrane phospholipids to form non-degradable complexes. The mechanism responsible for the inhibition of lysosomal phospholipases by CADs is unknown, although it has been suggested that it is reversible and not specific for a particular enzyme [71]. The dose-dependent inhibition of phospholipases A1, A2 and C has been reported in animal models exposed to amiodarone, chlorpromazine and chloroquine [72]. It has been observed that amiodarone, an antiarrhythmic extensively studied as a prototypical phospholipidosic drug, binds to hydrophobic moiety of phospholipids avoiding its degradation by phospholipases. In contrast, the nature of interactions of other drugs with phospholipids and their potential inhibition of phospholipase activities remain unknown. Lysosomal dysfunction by the release of lysosomal hydrolases associated with extracellular vesicles, impaired transport of lysosomal enzymes from endoplasmic reticulum or gene expression changes in enzymes involved in lipid metabolism have been also suggested as drug-induced phospholipidosis mechanisms [37,62,69]. An in vitro gene expression study revealed that amiodarone and other CADs induce multiple effects on proteins involved in lipidic metabolism, including the downregulation of phospholipase enzymes and lysosome enzyme transport and induction of phospholipid biosynthesis [37]. Accordingly, increased phospholipid synthesis was evidenced experimentally after exposure to amiodarone or chloroquine, suggesting that, in addition to impaired phospholipid degradation, enhanced phospholipid synthesis may contribute to drug-induced phospholipidosis [71,73]. However, it remains unclear if the observed effects on phospholipid biosynthesis are a compensatory response to maintain phospholipid levels in a situation of impaired lipid recycling via endosomal system [71]. In summary, phospholipidosis is a multifactorial phenomenon associated with prolonged exposure to drugs and no single mechanism is able to explain the effects triggered by all known phospholipidosis-inducing compounds. The consequences of the excessive accumulation of phospholipids and drugs in lysosomes are difficult to predict, as many lysosomal and non-lysosomal cell functions may be disturbed, but it is generally considered a reversible event with the progressive normalization of lysosomal function after drug withdrawal [68].

### 2.4. Drug-Induced Cholestasis

Cholestatic pattern is frequently associated with DILI and may appear in up to 50% of DILI cases [18,24,45]. Antimicrobials, NSAIDs, immunomodulators, antipsychotics, hypoglycemic drugs and steroids are the therapeutic groups mostly associated with cholestatic effects. Cholestasis is a multifactorial liver disorder characterized by the impairment of bile flow from liver to duodenum, which leads to intrahepatic accumulation of bile acids (BAs) and subsequent toxicity to liver cells. In most cases, bland cholestasis induced by drugs is reversible and patients recover completely after the cessation of the drug; however, in some patients, cholestatic liver injury may persist and intrahepatic bile ducts are progressively damaged and lost [22]. The resulting ductopenia can lead to the near complete destruction of ducts (vanishing bile duct syndrome) with inflammatory response directed at cholangiocytes, chronic cholestasis and biliary cirrhosis. The prognosis of patients with severe bile duct loss is generally unfavourable [19].

The functional alteration of hepatobiliary transport systems is considered a major mechanism of drug-induced cholestasis. The inhibition of BSEP (bile salt export pump), the major transporter responsible for the canalicular efflux of BA, has been documented for many cholestatic drugs (e.g., erythromycin, bosentan, cyclosporine, itraconazole and ethynyl estradiol) [74]. The interaction of drugs, or their metabolites, with other efflux transporters such as MDR3 (multi-drug resistance protein 3) (e.g., cyclosporine, itraconazole, haloperidol and chlorpromazine) and MRPs (multi-drug resistance-associated proteins 2, 3 and 4) (e.g., erythromycin, bosentan, cyclosporine and ethynyl estradiol), or with NTCP (sodium-dependent taurocholate co-transporting polypeptide) (e.g., cyclosporine and chlorpromazine), the major uptake system of BA from blood, have also been reported [74,75,76]. In addition to the direct inhibition of hepatobiliary transport systems, changes in the expression or localisation of transporters in the membranes of hepatocytes, disturbance of hepatocyte polarity and cytoskeleton alterations with the disruption of pulsatile bile flow have also been postulated to explain cholestatic consequences of DILI. Canalicular expression of BSEP, MDR3 and MRP2 is disturbed in patients with cholestatic DILI [77] and the ability of model cholestatic drugs (cyclosporine, chlorpromazine, bosentan, azathioprine and glyburide) to alter the expression of key basolateral and apical biliary transporter has been shown in an in vitro study [41]. In addition, drugs (e.g., ethynyl estradiol) can affect the trafficking of hepatobiliary transporters from the endosomal compartment to the biliary canalicular domain, promoting the internalization and degradation of transporters and altering the functional polarity of hepatocytes [42]. The expression of BA transporters in human liver shows a considerable interindividual variability and certain genetic polymorphisms in BA transporters might predispose to hepatic disorders, such as drug-induced cholestasis [76,78]. However, such genetic variants are not sufficient to explain most cases of idiosyncratic drug-induced cholestasis and additional susceptibility factors should be considered. In summary, although the mechanisms involved in drug-induced cholestasis have been extensively investigated, many aspects remain to be clarified, in particular, those related with its idiosyncratic nature [42,74,76].

## 3. Liver Cells and Their Role in Drug-Induced Toxicity

The liver has a complex cell structure and is composed by different cell types, including both hepatocytes or parenchymal cells and NCPs, such as liver sinusoidal endothelial cells (LSECs) (HSC, KC, immune cells and bile duct epithelial cells (or cholangiocytes) [13]. Hepatocytes, the most abundant liver cells, are polarized polygonal-shaped cells that perform the majority of the synthetic, metabolic and homeostatic functions of the liver [79]. Their particular configuration and distribution in the hepatic lobule allow the hepatocytes to present specific basolateral and apical domains that are essential for their role in the uptake, metabolism and biliary elimination of endogenous and exogenous molecules. The transporters located at the basolateral (sinusoidal) domain facilitate the entry of substances from blood, whereas apical transporters participate in the efflux of molecules into the bile canaliculi [76]. Hepatocytes are responsible of the synthesis of most circulating proteins (albumin, plasmatic transporters, coagulation factors, lipoproteins and inflammation mediators) and the homeostatic control of a diversity of compounds (glucose, triglycerides and cholesterol, BA, amino acids or vitamins) [14]. They are also key players in the detoxification of both endogenous (ammonium and bilirubin) and exogenous (drugs and other xenobiotics) compounds. In fact, the high expression of phase I and phase II biotransformation enzymes shown by hepatocytes allows the liver to play a determinant role in the metabolism and toxicity of drugs [50]. Although in most cases drugs are metabolized into biologically inactive metabolites (detoxification), some drugs can be transformed into reactive metabolites capable of interact irreversibly with cell macromolecules (bioactivation) [50]. This metabolic bioactivation is considered an undesirable event with a determinant role in drug-induced hepatotoxicity.

Hepatocytes alone are not able to support all hepatic processes and NPCs are essential to the maintenance of liver structure and are important contributors to liver functions. Furthermore, NPCs are actively involved in various mechanisms of drug-induced liver toxicity.

LSECs, the most abundant NPC in the liver, are specialized endothelial cells characterized by the presence of fenestrations in their plasma membrane allowing the diffusion of many substances between the blood and basolateral membrane of hepatocytes [79]. LSECs present a scavenging function and endocytic capacity to eliminate macromolecular waste products and to detect damage-associated molecular patterns (DAMPs) released from injured hepatocytes [13]. LSECs secrete cytokines and other signal molecules playing a role in the activation of immune cells during drug-induced hepatotoxicity. In addition, they express phase I and phase II biotransformation enzymes and may help hepatocytes to bioactivate drugs into hepatotoxic metabolites.

HSCs, located at the space of Dise between hepatocytes and LSECs, are quiescent cells that constitute the largest physiological store of retinoids (vitamin A) in the body [13]. In response to liver injury, HSCs acquire a profibrogenic phenotype characterized by the loss of retinoids and high production of a large amount of extracellular matrix (ECM) components (e.g., collagen, elastin and proteoglycans) [79]. In parallel to the activation process, HSCs undergo morphological changes from normal star-shaped cells to myofibroblastic-like cells. Activated HSCs are able to synthesize cytokines and growth factors and participate actively in the inflammatory and fibrotic response of the liver during chronic hepatotoxicity [13]. In particular, HSCs play a key role in the progression of steatosis to more severe liver pathologies, such as steatohepatitis or cirrhosis [80].

KCs, the resident macrophages in the liver, have a high endocytic/phagocytic capacity and play an important function in host defense against soluble bacterial products and endotoxins [79]. They actively participate in the regulation of liver homeostasis and are involved in several mechanisms of liver injury induced by drugs and other xenobiotics [81]. KCs have been related to idiosyncratic hepatotoxicity induced by certain drugs by promoting the release inflammatory cytokines, growth factors and ROS that can precipitate DILI and/or contribute to exacerbate the response mediated by other liver cells (hepatocytes and HSCs) or by recruited immune cells (infiltrating macrophages, neutrophils or natural killer cells) [82,83].

Cholangiocytes are the cells that form the bile duct epithelium. They are polarized cells with well-defined basolateral and apical membranes that express multiple receptors and channels that efficiently modify the composition of the bile [84]. Cholangiocytes are target cells in several liver diseases, including drug-induced cholestasis and vanishing bile duct syndrome [19]. Enhanced cell proliferation and apoptosis are among the characteristic responses of cholangiocytes during cholestatic liver injury. Although the mechanisms of drug-induced cholangiocyte damage have not been fully elucidated, studies in experimental models of drug-induced cholestasis suggest the release of proinflammatory cytokines by hepatocytes and NPCs, and increased ROS production as potential causes of cholangiocyte injury [84].

## 4. In Vitro Cell Models for Chronic Hepatotoxicity

There is an urgent need of human-relevant in vitro models for preclinical testing during drug development processes. Although single-cell-type 2D cultures are still the most used for hepatotoxicity assessments [85], DILI is a complex process in which different cell types may be involved. Additionally, these systems usually cannot be cultured over long periods, thus limiting the study of chronic effects. Different in vitro cell-based models presented in 2D or 3D configurations used for the evaluation of chronic DILI have been described to remove toxic compounds from the pharmaceutical development and to design safer molecules that would finally reduce the attrition rates in preclinical and clinical trials. Normal cell function and physiology depend on cell-to-cell and cell–extracellular matrix interactions in a 3D microenvironment. In order to better emulate the liver environment, different complex cell culture models have been developed. These models include sandwich cultures and more complex 3D models, such as cultures within hydrogels or scaffolds and organ-on-a-chip (OoC) platforms. Each cell model exhibits important advantages and limitations as it is explained along this section. For instance, PHHs allow directly examining liver’s patient and in 3D configuration can be maintained for long periods, although important interindividual variations can be also observed. Immortalized cell lines lack important hepatocyte functions, although their easy handling and availability have led to being widely used for hepatotoxicity assessments. HLCs derived from iPSCs enable unlimited proliferation and self-renewal and allow reflecting population variability; however, the differentiation protocols still lead to obtain an immature phenotype. The 3D co-cultures of hepatocytes and NPCs is probably the system that better represents the environment and phenotype of the liver and that it is closer to in vivo conditions; however, they present still many limitations mainly due to the difficulty to expand cells in vitro, their random distribution and the difficulties to keep their phenotypes for longer periods [85].

Table 2 exemplifies different liver in vitro cell models for assessing chronic hepatotoxicity.

**Table 2 ijms-23-11428-t002:** Cellular models for studying chronic hepatotoxicity.

Cellular Model	Long-Term Stability	Characterization	Hepatotoxicity Assessments	Ref.
**2D cultures**				
PHH collagen (serum free conditions)	Up to 4 weeks	- Maintenance of functionality (urea and albumin production, CYP activities), morphology and expression of hepatic markers	- 6 drugs (cyclophosphamide, acrolein, rifampicin, loratidine, atorvastatin, APAP) for acute (48 h) or chronic (23 days) exposure	[86]
PHH sandwich	Up to 2 weeks	- Maintenance of functionality (urea and albumin production, CYP activities), polarisation and expression of hepatic markers up to 7–10 days	- Cholestatic compounds (+/−BA) incubated for 24 h. Assessment of CIx and urea production - Repeated dose of 7 model compounds (14 days). Measurement of ATP content	[87] [88]
HepaRG cells	Up to 4 weeks (+2 weeks-treatment)	- Maintenance of phase I and phase II enzymes activity and expression levels. - Stable levels of antioxidant enzymes - Maintenance of morphology	- Assessment of drug-induced steatosis after acute or chronic exposure) - Long-term repeated dose up to 14 days. Evaluation by HCS	[89,90] [91,92,93,94]
Upcyte Human Hepatocytes	Up to 3 weeks	- Maintenance of expression and activity levels of drug-metabolizing enzymes - Stable levels of antioxidant enzymes	- Long-term repeated dose up to 21 days. Evaluation by HCS (mechanistic studies including drug-induced phospholipidosis and steatosis)	[92,95]
HLC	Up to 2 weeks	- Commercially available HLCs - Maintenance of key hepatic markers	- Repeated dose (2, 7, or 14 days) of 4 model compounds (amiodarone, aflatoxin B, troglitazone and ximelagatran). - Drug-induced phospholipidosis and steatosis	[96] [96,97]
**3D cultures**				
PHH spheroids	Up to 5 weeks	- Maintenance of expression and activity levels of drug-metabolizing enzymes - Proteomic analysis revealed resemblance with liver in vivo.	- 123 drugs (14-day repeated-dose exposure). ATP content. - Fialuridine (32-day repeated-dose exposure) Changes in ATP, viability, lipid content and ROS production - Drug-induced cholestasis (cholestatic compounds +/BA) at repeated-dose exposure (up to 28 days)	[98] [99] [100]
HepG2 spheroids	Up to 1 week	- Histological characterization, CYP activities, albumin secretion, expression hepatic markers and CLF (transport)	- Repeated-dose exposure (6 days)	[101]
HepaRG Spheroids	>4 weeks	- Higher levels of liver-specific genes involved in drug metabolism, BA transport, and energetic pathways and secreted more albumin, glucose, and urea compared to those in 2D cultures	- 7-day exposure to model compounds. Multiplex hepatotoxicity and CYP induction assay. - 14-day repeated-dose study - Evaluation of drug-induced cholestasis and BA effects.	[102] [103] [100,103]
HLC spheroids		- RNAseq and functional profiling. Differentiation of immature vs. mature zonal hepatocytes - BA production and transport	- Screening of 238 compounds (24 h). Measurement of viability and CLF uptake, and mitochondrial-induced toxicity - Modeling CYP2C9*2 iPSC-liver organoids and susceptibility to bosentan-induced cholestasis	[104]
**Co-cultures**				
Micropatterned co-cultures fibroblast + PHH	Up to 4 weeks	- Secretion of urea and albumin, functional bile canaliculi; metabolisation of compounds using active Phase I and Phase II drug metabolism enzymes	- 45 drugs (14 days exposure, 4 concentrations up to 100× C_max_). - Evaluation of urea, albumin and GSH.	[105]
Micropatterned co-cultures HLC+ fibroblasts	Up to 4 weeks	- Improved functionality (polarity, albumin and urea secretion, Phase I and II activities, induction, down+-regulation of fetal markers)	- 47 drugs (6-day treatment) Evaluation of albumin, urea, ATP	[106]
3D scaffold (PHH, stellate, KC and endothelial cells)	Up to 3 months	- Maintenance of the production of albumin, fibrinogen, transferrin and urea; CYP inducibility, bile canaliculi-like structures and response to inflammatory stimuli	- 3 and 15 days (troglitazone, APAP, trovafloxacin). LDH release.	[107]
3D InsightTM Human Liver Microtissues (PHHs, endothelial, KCs)	Up to 5 weeks	- Morphological characterization, glycogen accumulation, polarized expression of transporters (BSEP, MDR1). Responsive to LPS treatment	- APAP, diclofenac and (14 days treatment (3 re-dosing) - Trovafloxacin +/LPS	[108]
Bioprinted 3D Primary liver tissues (human stellate cells, HUVECs, PHHs)	Up to 4 weeks	- Maintenance of ATP levels, albumin as well as expression and drug-induced CYPs. In vivo relevant architecture	- Trovafloxacin and levofloxacin (repeated-dose 7 days). Evaluation of albumin and ATP content. Analysis of histological effects	[109]
**Organ-on-a-Chip Platforms**				
Liver on a chip: bioprinted HepG2 spheroids	Up to 4 weeks	- GelMa hydrogels containing HepG2/C3A maintained the expression of key hepatic markers. Monitoring of biomarkers.	- Toxicity APAP (6 days). Monitoring levels of albumin, A1AT, transferrin and ceruloplasmin.	[110]
Liver-Chip (PHH KC + endothelial)	Up to 2 weeks	- Drug-metabolising capacity, albumin secretion, transporters functionality	- Toxicity of APAP (7 days) Human and cross-species toxicities - Methotrexateand fialuridineinduced fibrosis-steatosis (7 and 10 days, respectively)	[111]
Biomimetic array chip (collagen 3D PHHs)	Up to 12 days	- Improved and stabilized liver functionality (viability, albumin and urea production). CYP functionality.	- Toxicity of 122 clinical drugs (treatment for 7 and 14 days). ATP content.	[112]
Multi-organ-chip (PHHs + stellate + skin)	Up to 4 weeks	- Expression of key hepatic markers. Maintenance of metabolic activities.	- Toxicity of troglitazone (6 days). Cell viability and transcriptomics analysis.	[113]

### 4.1. Primary Human Hepatocytes (PHHs)

PHHs are considered the gold standard in vitro model for testing drug-induced hepatotoxicity during drug development because they remain functional for 24–72 h, allow medium-throughput screening and reflect inter-individual differences in metabolism [9,114]. PHHs have been widely used for evaluating not just viability but for examining different pre-lethal mechanistic effects, such as steatosis, cholestasis, phospholipidosis, mitochondria membrane function or oxidative stress [9,115,116]. However, the scarcity of liver tissues for isolating PHHs as well as their phenotypic instability and the short lifespan limit their application for DILI evaluations, particularly for long-term studies, or for reproducing the pharmacokinetics of drug exposure. Recently, different strategies for maintaining hepatocyte function in vitro for a long time, such as the use of a small-molecule combination [117] or their culture in serum-free conditions [86], have been proposed for keeping the PHH phenotype up to 4 weeks, which would allow their use for long-term hepatotoxicity assessments. In fact, PHHs cultured under serum-free conditions have been used for studying the acute (48 h) and chronic (23 days) toxicity of six drugs, validating the sensitivity of the system for long-term assessments. Significant changes in albumin secretion and CYP3A4 activity were found after 4 weeks treatment with atorvastatin or acetaminophen (APAP), even at subcytotoxic concentrations, indicating their suitability for long-term studies [86]. This is important because, although APAP is known to induce acute DILI, this study demonstrated that PHHs, under the described conditions, can be exposed to drugs for long periods (4 weeks), and, in order to simulate in vivo drug metabolism and hepatotoxicity, in vitro models should keep morphology and identity, including drug response. On the other hand, although PHHs are the gold standard for Hepatology studies, they only reflect the responses and effects on hepatocytes, whereas the liver contains other cell types, such as NPCs, which also contribute to the development of liver disease. Thus, advanced in vitro models try to also incorporate other cells, such as KCs, and study their contribution to chronic DILI [118,119].

### 4.2. PHHs Cultured in Sandwich Configuration

In order to keep liver functionality over longer periods, a sandwich configuration has been extensively used. In this system, PHHs are cultured between two matrix layers, which allows the development of polarized cell surface domains over several days in culture, and the formation of functional basolateral and apical domains that mediate cell uptake and biliary efflux, respectively [9,120]. Additionally, the culture on Matrigel or collagen sandwich configuration improves the secretion of urea and albumin and augments both basal and induced drug-metabolizing enzyme activities [14,120,121].

Sandwich-cultured hepatocytes in combination with a concentrated mixture of BA have been used to identify compounds provoking cholestasis by changing BA composition [87]. In this study, PHHs cultured on collagen sandwich configuration were incubated with cholestatic drugs (cyclosporin A, troglitazone, chlorpromazine, bosentan, ticlopidine, ritonavir and midecamycin) and/or compounds causing hepatotoxicity by other mechanisms (diclofenac, valproic acid, amiodarone and APAP) in the presence or absence of BA. Cholestatic compounds showed increased toxicity in the presence of BA, while no significant changes were observed for the other compounds [87].

On the other hand, PHHs cultured in sandwich configuration have been also used for long-term hepatotoxicity assessments, although, compared to PHH spheroids, they exhibited lower predictivity and a loss of key hepatic cell functions [88].

### 4.3. HepaRG Cells

HepaRG cells are currently considered the most promising cell line for hepatotoxicity assessments. Proliferating cells show bipotentiality features; thus, cells are able to differentiate into a mixed population of hepatocyte-like and biliary-like cells [122]. After differentiation into a hepatocyte phenotype, they exhibit higher levels of phase I and phase II drug-metabolizing enzymes than any other hepatoma cell line [123]. Additionally, HepaRG cells express hepatobiliary transporters at similar levels to PHHs, which is important for the in vitro evaluation of drug-induced cholestasis [124]. Moreover, differentiated HepaRG cells keep their metabolic competence for several weeks, which allows performing long-term repeated-dose studies [125]. Under serum-free conditions, HepaRG exhibited urea and albumin production at similar levels to the in vivo situation, although CYP activities levels declined with time in culture [125]. In addition to chronic hepatotoxicity, steatosis can be induced in vitro in HepaRG [90,126], which suggests them as an attractive cell model to analyze changes in lipid metabolism due to drug-induced steatosis. Finally, although they do not consider population differences [91,93,127], they have been widely used as a model for studying chronic hepatotoxicity.

Several authors have used differentiated HepaRG for long-term toxicity evaluations for a maximum of 14 days after differentiation at clinically relevant concentrations [91,92,93,94]. The use of HepaRG cells following repeated-dose regimes allow using lower concentrations more relevant to the clinics and obtaining better predictions. HepaRG cells have been also used for longer periods of treatments, although a distinct strategy was used in these cases since the complete differentiation process was unfinished [128,129]. For instance, HepaRG cells were treated for 3 weeks with a range of concentrations of bisphenol A, beginning at the proliferating period in order to mimic liver development [128]. Bisphenol A induced steatosis, with significant effects on triglycerides and neutral lipids, although the expression of key lipid and carbohydrate metabolism was unaltered [128].

### 4.4. Upcyte Human Hepatocytes

In recent years, Upcyte human hepatocytes (UHHs) have been used for both acute and repeated dose hepatotoxicity assessments [95,130]. UHHs are immortalized hepatocytes that are able to proliferate while keeping key hepatic functions for up to 3 weeks [95]. The most important advantage of UHHs is the unlimited availability of hepatocytes from different donors, which permits reflecting the variability among the population.

UHHs have been shown to maintain the expression and activity levels of drug-metabolizing enzymes up to 3 weeks [95]. UHHs were used for the investigation of long-term hepatotoxicity of 15 drugs (11 hepatotoxic and 4 non-hepatotoxic) after a repeated-dose strategy at clinically relevant concentrations. The study was based on the use of high-content screening (HCS) technology that allows the detection of the underlying mechanisms of drug-induced toxicity before cytotoxicity occurs [95], and revealed the suitability of UHHs for long-term repeated-dose studies that can detect the potential hepatotoxicity in preclinical development. Recently, we have also demonstrated that UHHs maintained the activity of important antioxidant enzymes and allow studying different mechanisms implicated in DILI, including oxidative stress, at sub-lethal concentrations [92]. In this case, UHH donors seemed to show a higher sensitivity than HepaRG detecting oxidative stress induced by test compounds and also provided information about differential responses in the distinct donors [92]. The importance of this study is that cells were exposed to concentrations around the therapeutic peak plasmatic concentration (C_max_), which is close to what happens in vivo. Even though UHHs can be maintained over longer periods, more complex models that incorporate 3D configuration or other cell types would be desirable for evaluating the response of the liver to the chronic exposure to drugs.

### 4.5. Hepatocyte-like Cells Derived from Pluripotent Stem Cells

Pluripotent stem cells (PSCs) include both embryonic and induced pluripotent stem cells (iPSCs) and can differentiate into all cell types of the body, while maintaining genetic stability. PSCs proliferate extensively in vitro and can be differentiated into hepatocyte-like cells (HLCs), which offers a stable source of hepatocytes for hepatotoxicity screening. Another important advantage is the fact that iPSCs also have the potential to establish genotype-specific cells from different individuals [131,132]. Since it has been reported that HLCs from different donors displayed inter-individual variations in CYP activities and in the expression of phase II drug-metabolizing enzymes and transporters as occurs in PHH [133,134], HLCs illustrating particular phenotypes of interest for DILI studies could be generated.

The potential of patient-specific iPSC-derived HLCs to model iDILI is also a major advantage of this system. For instance, it has been demonstrated that pazopanib, a tyrosine kinase inhibitor used for advanced renal carcinoma, was more toxic to HLCs derived from subjects that showed previous episodes of DILI with this specific drug, illustrating their value to broadly study the potential risk factors in susceptible patients [135].

On the other hand, the iPSC technology has been also used to generate HLC-derived from patients with BSEP deficiency. The cell model developed mimicked the pathophysiology of familial intrahepatic cholestasis type 2, characterized by an impaired biliary excretion and reduced expression of BSEP, which were rescued by drug treatment [136].

Another important advantage of HLCs is that can be maintained in culture for longer periods, allowing repeated-dose studies. Holmgren et al. (2014) demonstrated that HLCs became more sensitive to amiodarone, aflatoxin B1 and troglitazone, after extended exposure (14 days). Mechanistically, the system also allows detecting drug-induced steatosis and phospholipidosis [96]. In this sense, other authors have also used HLCs for studying drug-induced steatosis and phospholipidosis [97], although after acute exposure. HLCs derived from iPSCs exposed to drugs known to cause hepatotoxicity through steatosis and phospholipidosis showed a similar response than HepG2 cells when different parameters, such as lipid and phospholipids accumulation, ROS production or changes in mitochondrial membrane potential were assessed by HCS [97].

The production of mature HLCs still needs to be improved since liver functions, such as CYP activities, remain low [137] and it is difficult to obtain a large number of cells. Some improvements to achieve a phenotype closer to the in vivo situation have been addressed in 3D cultures and in combination with other cell types.

### 4.6. Three-Dimensional Cultures

Spheroids are aggregates of cells that typically form a spheroidal structure in the presence or absence of a scaffold and that can include one or more cell types [138]. They recapitulate the 3D environment that occurs in vivo and allow culturing cells for longer periods. From a toxicological point of view, since certain toxicities involve different cell types, multicellular spheroids may allow detecting these toxicities although the increased complexity also leads to more complex data analysis and interpretation.

#### 4.6.1. HepG2 Spheroids

A 3D in vitro model using Matrigel hydrogels for the long-term culture of HepG2 cells has been described [101]. In this model, HepG2 cells show many phenotypic characteristics of hepatocytes in vivo, such as reduced proliferation, polarization and increased expression of albumin, urea, xenobiotic transcription factors, phase I and II drug metabolism enzymes and transporters. The use of this model in a repeated-dose scenario (up to 6 days) revealed increased sensitivity in identifying hepatotoxic compounds, and, since the cells can be maintained up to 4 weeks, longer treatments could be applied, using more relevant concentrations [101]. Although HepG2 cells have been widely used for hepatotoxicity studies, one of their major limitations is the low phase I activity levels, which would not recapitulate in vivo physiology and would render them only acceptable for the evaluation of non-bioactivable drugs. To face this limitation, the transfection of HepG2 cells with adenovirus encoding for the most relevant CYPs has been proposed to create metabolically competent HepG2 cells [139].

#### 4.6.2. HepaRG Spheroids

Different 3D HepaRG models have been described to show a better performance than 2D cultures [140]. Differentiated HepaRG spheroids show high levels of CYP activities, polarized hepatocyte architecture and functionality up to 4 weeks. HepaRG spheroids have been widely used for acute and chronic hepatotoxicity studies and for metabolism studies [102,103,141]. Undifferentiated HepaRG cells encapsulated in collagen hydrogels under DMSO-free conditions have been shown to form polarized spheroids of differentiated hepatocytes with high levels of expression and activity of drug-metabolizing enzymes [103]. This model resulted to be useful for studying DILI after a 14-day repeated-dose exposure regime. Additionally, compounds that have been shown to produce cholestasis (cyclosporin A, bosentan, amiodarone and chlorpromazine) showed synergistic toxicity when incubated in the presence of BA [103]. The model also demonstrated to be suitable for steatotic stimulation, mutagenesis and genotoxicity studies [103]. This study used concentrations in the range of 1–100 C_max_, which is close to the in vivo situation and corroborated the sensitivity to induce liver injury.

#### 4.6.3. PHH Spheroids

Bell et al. (2016) developed a 3D PHH spheroid system in serum-free conditions that maintained morphology, viability and functionality up to 5 weeks [142]. Additionally, the system allowed the co-culture with other NPCs, which could offer important mechanistic information. Interestingly, the fact of using PHHs permitted the maintenance of inter-individual variability, which is important for the study of iDILI. The authors demonstrated the suitability of the test system for studying long-term hepatotoxicity by exposing the PHH spheroids to five different hepatotoxins (amiodarone, bosentan, diclofenac, fialuridine and tolcapone) following a repeated-dose exposure regime. For all hepatotoxic compounds, prolonged exposure resulted in increased toxicity, which allows to detect the toxicity in vitro at clinically relevant concentrations [142]. On the other hand, Vorrink et al. (2018) used the PHH spheroid system for screening a large set of compounds (123 drugs) after chronic exposure (14 days) and showed 100% specificity and 69% sensitivity at clinically relevant concentrations [98]. PHHs have been also used for the evaluation of drug-induced cholestasis, analyzing the synergistic effects of BA [100].

#### 4.6.4. HLC Spheroids/Organoids

Human liver organoids derived from PSCs have shown to express typical hepatic markers and increased expressions of genes related to polarity and transporters activities [104]. Although the organoids were immature, they allow the identification of compounds with toxic potential. The authors incubated these organoids with more than 200 toxic compounds and combined the measurement of viability and CLF uptake for determining their toxicity. Additionally, the use of HLC organoids derived from different donors also allowed studying the differential response of donors with genetic predisposition (CYP2C9*) to bosentan-induced cholestasis, as seen in the clinics [104]. To determine the predictability of the systems, the authors analyzed their toxicity data at the concentration closest to the C_max_ value, obtaining good sensitivity and specificity. Interestingly, the test model enables the detection of indomethacin-induced toxicity at concentrations close to the plasmatic concentration, and, thus, to the in vivo situation.

### 4.7. Co-Cultures

Co-culture of PHHs with liver or non-liver cells has been shown to stabilize and improve the phenotype of PHHs [143]. The co-cultures of PHHs and NPCs have been shown to increase viability and functionality, and also allow to investigate the contribution of the inflammatory mechanisms to toxicity [107]. The involvement of pro-inflammatory cytokines and the interactions between KCs and hepatocytes provide a physiologically relevant model [118] that has been also proposed for hepatotoxicity studies. Immune cells have a critical role in both the increase and reduction in inflammation, and, thus, in the progression of immune-mediated DILI. Therefore, it is essential to understand the interactions between immune cells and other cells in the liver [119]. In order to minimize the complexity of the models, some systems have used cytokines or conditioned media from immune cells [82,144]; however, in order to elucidate the multifaceted crosstalk between hepatocytes and immune cells, more complex models are required. The majority of in vitro co-cultures have focused on the combination of PHHs and KCs in a variety of culture formats [119]. In some studies, a ratio of 10:1 (PHH:KC) was used, although other authors have tried to represent the inflammatory conditions by using a 2.5:1 ratio [118,119]. Three-dimensional InsightTM Human Liver Microtissues that contain PHHs and NPCs (KCs and endothelial cells) are a commercially available system (InSphero, Schlieren, Switzerland) that can be maintained up to 5 weeks in culture with good viability and functionality and have been used for the assessment of chronic toxicity and inflammation-mediated toxicity [108]. The induction of the inflammatory response by lipopolysaccharide (LPS) resulted in increased levels of IL-6 and increased sensitivity to trovafloxacin, confirming the importance of adding NPCs for studying immune-mediated DILI. Jiang et al. (2019) used the same system containing PHHs and KCs to study APAP-induced toxicity in the presence of LPS. The presence of LPS exacerbated APAP toxicity and was also associated with disrupted mitochondrial function and elevated production of IL-8 and suppressed production of IL-6 [145], indicating their suitability for studying inflammation-associated drug toxicity, although drugs to be known to produce chronic DILI should be explored to validate the system.

A multi-well insert plate system (RegeneMed, San Diego, CA, USA) was used to culture PHHs and NPCs (KCs, endothelial and hepatic stellate cells) for up to 3 months, maintaining key hepatic functions, such as the production of urea, albumin or fibrinogen, CYP activities and the formation of bile canaliculi structures [107], similar to in vivo situation. These 3D liver cell cultures also showed more closely in vivo responses to LPS during inflammation than monolayers of PHHs. Finally, this test system was then incubated with seven different model compounds for up to 15 days at concentrations around the in vivo plasma concentration on a daily basis, and allow a better DILI detection, including species-specific drug effects, than monolayer hepatocyte cultures [107]. The presence of NPCs turned out to increase the sensitivity for the detection of drug-induced hepatotoxicity with an inflammatory component, for example, triggered by KCs since they probably better reflect the physiological conditions [107].

In order to avoid the scarcity of fresh tissue to isolate both PHHs and NPCs, such as KCs, some authors have used alternative cell sources, such as HepG2 and THP-1 monocytes/macrophages, for co-culturing purposes [83,146]. The co-cultures of HepG2 cells and THP-1 macrophages showed increased toxicity to troglitazone, trovafloxacin, diclofenac and ketoconazole in the presence of TNF-α or LPS, confirming that an inflammatory microenvironment enhances the sensitivity of liver cells towards iDILI drugs [83].

Finally, other authors have proposed the co-culture of PHHs or HLCs with mouse fibroblasts for enhancing hepatocyte functionality [105,106,147]. These micropatterned co-cultures stabilize the phenotype of both PHHs and HLCs derived from iPSCs, allowing also their used for long-term hepatotoxicity assessments with a better predictivity than mono-cultures [105,106,148].

### 4.8. Microfluidic Liver-on-a-Chip Systems

OoC technology recreates 3D organ microenvironments, such as multicellular architecture, vascular perfusion, fluid flow and other relevant physiological microenvironment of organs [111]. Microfluidic devices can be used to subject cells to shear forces and gradients as occurs in vivo as well as allowing the automated delivery of nutrients [5]. Previously, the OoC field was focused on the design and characterization of these microfluidic devices; however, at present, the challenge is to prove their superiority to animal models.

In addition to the beneficial effects of perfusion on the functionality of liver cells, it also permits to subject the cells to gradients of oxygen, nutrients and hormones that lead to liver zonation or differential functions in hepatocytes across the sinusoid [149]. For instance, Allen et al. studied zonal hepatotoxicity by perfusing APAP for 24 h and showed increased cell death at the low-oxygen region similar to centrilobular necrotic patterns observed in vivo [150]. Thus, advance bioreactor systems that can modulate oxygen and nutrients could lead to a better understanding of DILI. Even though APAP is a model for acute DILI, the study demonstrated that microfluidic systems allow to study differential responses depending on the oxygen that the cells receive and allowing the study of differential hepatotoxicity. Chronic drugs should be now assessed to study how microfluidics help in the mechanistic understanding of DILI.

#### 4.8.1. Liver-on-a-Chip Platforms

Bhise et al. (2016) described a liver-on-a-chip platform that allowed bioprinting of HepG2/C3A spheroid-laden hydrogel constructs directly within the culture chamber of the bioreactor [110]. The system also enabled the monitoring of serum biomarkers, such as albumin or α1-anti-trypsin, and the culture up to 30 days. The exposure of this tests system to APAP for 24 h resulted in a decrease in the metabolic performance and serum biomarkers, confirming its suitability for drug toxicity analysis [110]. This technology has been also used for screening large sets of compounds after repeated-dose exposure with high selectivity and sensitivity [112]. The system described by Xiao et al. showed an ureogenic capacity comparable to native in vivo production and used concentrations of toxicants up to 100× C_max_, and the toxicity values were used as thresholds to differentiate between hepatotoxic and non-hepatotoxic drugs, which is important for establishing the clinical relevance of the model [112].

Microfluidic systems using co-cultures with NPCs have been extensively described, even for hepatotoxicity assessments, although they have been mainly used for evaluating acute [151,152] and not chronic hepatotoxicity. Since the importance of NPCs is essential for the understanding and prediction of chronic DILI, in future years, it is expected to have more complex models that allow evaluating the role of each cell type in the development of chronic toxicity.

#### 4.8.2. Multi-Organ-on-a-Chip Platforms

In order to fully address systemic drug effects, multi-organ-on-a-chip platforms with multiple interconnected tissues have been proposed. These in vitro platforms recapitulate aspects of the in vivo crosstalk between different tissues at a miniaturized scale. For instance, recently, intestinal and liver platforms have been connected to model first pass drug metabolism and intestine liver interactions [5]. These systems can display tissue specific functions up to 2 weeks [153]; therefore, they could be used in long-term DILI studies. Commercial organ-on-a-chip platforms are also available (Mimetas, Hesperos and Draper) [5].

On the other hand, the heart is another major organ of interest in risk assessment. Moreover, drug metabolites from the liver can influence cardiotoxicity and have led researchers to explore heart–liver models for evaluating their interactions. Lee-Montiel et al. (2021) have recently developed a microphysiological system that uses the same iPSC differentiated into cardiomiocytes and HLCs for studying drug interactions that could also be applied for studying chronic DILI [154].

Although different combinations of multi-organ-on-a-chip have been proposed, their application for chronic hepatotoxicity assessments is very limited. Wagner et al. (2013) used a dynamic multi-organ chip composed of human artificial liver microtissues (HepaRG + human hepatic stellate cells) and skin biopsies for the evaluation of chronic exposure (6 days) to troglitazone [113]. The co-cultures showed long-term performance in the microfluidic multi-organ-on-chip device up to 4 weeks, and showed a dose-dependent response to troglitazone, indicating their suitability for long-term evaluations [113].

## 5. Tools for Assessing Chronic DILI

To assess DILI potential, it is common to assess general toxicity markers, such as lactate deshydrogenase or ATP release. However, since there are multiple mechanisms implicated in DILI, the application of readout multiparametric technologies, such as HCS, transcriptomics, metabolomics or proteomics that provide information at distinct levels, have been proposed.

Fluorescent probes and protein-based fluorescent sensors are widely used for live cell studies. HCS enables combining numerous fluorescent probes that are indicators of viability, apoptosis, ROS production or mitochondria markers [155,156]. This technique can be used in more complex multicellular 3D models, which usually requires the use of higher magnifications, confocal imaging and 3D analysis [157].

Among the omics techniques, transcriptomics includes studies of mRNA transcripts, miRNA and DNA methylation patterns, whereas proteomics is used to discover changes in protein expression. Metabolomics evaluates changes in the metabolites secreted into the culture medium or in the cells. Table 3 summarizes different proposed assays for evaluating chronic hepatotoxicity, differentiating by mechanism.

### 5.1. Techniques for Studying Autoimmune DILI

To further examine the role of inflammation immune-mediated DILI, the synergistic effects of the addition of LPS or different combinations of cytokines and the incubation of drugs have been widely explored [82]. Using this strategy, the drug–cytokine synergistic induction of toxicity to hepatocytes or HLCs was observed [82,158,159]. In addition, not only was direct drug-induced hepatocellular death recapitulated in vitro, but also immune cell-mediated secondary hepatotoxicity [158]. Transcriptomics and cytokine profiling and the analysis of the activation of caspases and protein kinases have been also used for more specifically assessed immune DILI [160,161]. Phosphoproteomic data suggested multiple intracellular signaling pathways involved in the response of hepatocytes to certain hepatotoxic drugs and inflammatory cytokines [160].

### 5.2. Techniques for Studying Drug-Induced Cholestasis

For the assessment of drug-induced cholestasis, the synergistic effects of BA have been tested to calculate the so-called cholestatic index (CIx) [162]. CIx values reflect the viability or functionality of the hepatic test systems exposed to a cholestatic drug and a 50× concentrated BA mix compared to treatment with the cholestatic drug alone [163]. Additionally, viability measurement, transcriptomics analysis of transporters (BSEP, MRP2 and NTCP) and enzymes implicated in BA metabolism (i.e., CYP7A1) have been commonly assessed. MS has been also used for measuring changes in BA profiling [164] as well as metabolomic changes produced by cholestatic drugs [165].

In order to measure the functionality of BA transport, different fluorescent dyes have been used. Typically, 5- (and 6)-carboxy-2-, 7-dichlorofluoresceindiacetate (CDF-DA) has been used as a model compound to characterize transporter-mediated biliary excretion function [87]. On the other hand, BA accumulation has also been evaluated with fluorescently labeled taurocholic acid derivatives [100,142].

### 5.3. Techniques for Studying Drug-Induced Phospholipidosis

In order to determine the phospholipidosic potential of drugs, distinct fluorescent probes to examine phospholipid accumulation by microscopy, flow cytometry or espectrofluorimetry have been proposed [62].

Sawada et al. (2005) proposed 12 gene markers of drug-induced phospholipidosis, suggesting that alterations in lysosomal function and cholesterol metabolism were involved in the induction of phospholipidosis. These genes were indicative of the inhibition of lysosomal phospholipase activity, the inhibition of lysosomal enzyme transport, increased phospholipidosis and cholesterol synthesis, pointing out key metabolic changes implicated in phospholipidosis development [37]. Finally, metabolomics, which enables the measurement of metabolites, has been proposed for detecting drug-induced phospholipidosis. For instance, decreased lisophospholipids to phospholipids ratio, as a result of phospholipid degradation inhibition, has been suggested as a fingerprint of drug-induced phopholipidosis [166,167].

### 5.4. Techniques for Studying Drug-Induced Steatosis

Different in vitro assays to assess the steatogenic potential of drugs have been proposed [62]. Drug-induced steatosis produced significant changes at mRNA level [168] and it is also characterized by modifications in different molecules, such as FFAs and other lipids, proteins and metabolites [169]. The simplest assays are based on the use of fluorescent probes, such as Nile Red or BODIPY, to determine neutral lipid accumulation. In recent years, the use of HCS has been incorporated to assess multiple mechanisms and changes in in vitro models. It has been demonstrated that the combination of different probes to measure not only lipid accumulation but also changes in ROS production or mitochondrial damage, better predict the risk of liver damage [170].

Transcriptomics analysis has revealed a common signature for drug-induced steatosis [90,126,171]. Steatogenic drugs have been described to produce changes in the expression profile of the genes that control lipid homeostasis, such as CROT, EHHADH, MTTP and ANGPTL3. Additionally, different in vivo and in vitro studies have shown compound-specific effects at the level of different lipid-related genes and transcription factors, which resulted in individual transcriptomics patterns [90,172,173,174]. In recent years, the miRNA profiles of drug-induced steatosis have been also defined [175].

Finally, the use of metabolomics has shown changes in the levels of lipids, such as ceramides and triglycerides, and carnitine in different cellular models of steatosis [176] that could be also used as biomarkers.

**Table 3 ijms-23-11428-t003:** Techniques used to specifically assess chronic hepatotoxicity.

Type of Injury/Technique	Markers	In Vitro Model	Ref.
Autoimmune DILI			
Transcriptomics	- Cytokine production (IL1b and IL89)	PHH	[177]
Cytokine profile	- Determination of proinflammatory cytokines (ELISA and Luminex)	3D microtissues, PHH	[145]
Protein Phospholyration	- Synergictic effects (LPS or cytokines + drugs)	PHH	[160]
Steatosis			
Fluorimetric assays	- Nile Red, BODIPY and Oil Red O	HepG2, PHH, HepaRG	[62]
HCS	- Lipids staining (BODIPY) + other makers (TMRM, DCF and viablity)	UHH, HepaRG, HepG2, HLC	[95,126,170]
Transcriptomics	- FA oxidation and transport (APOB and ACADL) - De novo lipogenesis (PPARG and THRSP) - CYPs - Transcription factors (FOXA1, HEX and SREBP1C) - miRNA	HepaRG HepG2 HepG2	[90,126] [171] [175]
Metabolomics	- Diacylglycerol and triglyceride accumulation and carnitine deficiency	HepaRG	[176]
Phospholipidosis			
Fluorescent probes	- NBD-PC, NBD-PE and LipidTox	HepG2, PHH, HepaRG, spheroids	[62,64]
HCS	- LipidTox + other probes	UHH, HLC	[95]
Transcriptomics	- Lysosomal phospholipase activity (ASAH1 and SMPDL3A) - Lysosomal enzyme transport (AP1S1) - Phospholipid biosynthesis (ELOVL6 and SCD) - Cholesterol biosynthesis (HMGCS1, HMGCR, DHCR7 and LSS)	HepG2	[37]
Metabolomics	- Levels of phosphatidylcholines, phosphatidylethanolamines, phosphatidylserines and phosphatidylinositols	HepG2	[166,167]
Cholestasis			
Transcriptomics	- Transporters (BSEP and NTCP) - Oxidative stress (NFR2 and GST) - Inflammation (IL1R1 and JUN) - ER stress (ATF4, ATF6 and DD1T3)	HepaRG, PHH spheroids, HepaRG spheroids	[100,163]
Cholestatic index	- Study of the synergistic effects of BA	HepaRG, PHH spheroids	[100,163]
Mass spectrometry	- BA profiling (intracellular and in the medium) - Metabolomic alterations (carnitine, ceramide and triglyceride accumulation)	PHH, HepaRG HepaRG	[164] [165]
Fluorescent dyes	- CDF-DA (transporter-mediated biliary excretion function) - Fluorescent-labelled BA	Sandwich-PHH, HLC PHH spheroids	[87,104] [100,142]

## 6. Conclusions and Future Perspectives

In recent years, several in vitro systems have been developed for toxicological applications. Regulatory boards strongly promote the application of NAMs for increasing the predictivity of DILI. In this sense, different companies (i.e., InSphero and HUREL) have proposed new cellular methods for predictive toxicology in the drug development processes. However, in order to validate and compare them, the viability and functionality of the models should be deeply analyzed. Among the available liver cell models, those reflecting the variability and heterogeneity of human population are of particular interest. PHHs, UHHs and iPSC-derived HLCs can be obtained from different donors and their application to toxicity studies may enable more accurate DILI predictions as well as the analysis of host-dependent factors (e.g., genetic background, age, sex and race) contributing to more severe or chronic episodes of liver damage. In addition, most of the systems usually are tested with a single drug or just few compounds. Thus, before industry can adopt these new platforms, extensive characterization and testing of larger panels of well-known hepatotoxicants that act through different mechanisms of action are required.

Currently, there is a vast amount of evidence that when cultured in 3D hepatic cell models show a better performance (higher albumin and urea secretion, higher phase I–III activities) and can be maintained longer than when in 2D cultures [138]. This is essential for applying repeated-dose regimes and for detecting chronic toxicities, although 3D cultures can be less reproducible than monolayers. On the other hand, culture models that contain a single cell type are unlike to recapitulate chronic DILI; thus, much progress is being made to develop strategies that consider multiple cell types, such as NPCs. Finally, instead of assessing a single endpoint involved in DILI, new testing strategies that provide mechanistic information have demonstrated to be more predictive and to improve the understanding of toxic effects.

## Figures and Tables

**Figure 1 ijms-23-11428-f001:**
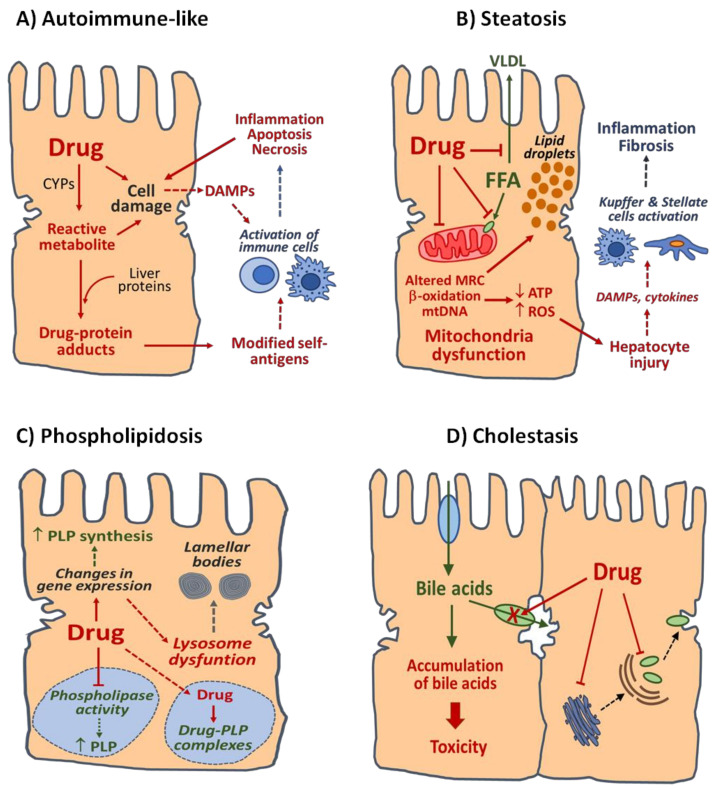
**Mechanisms of four main types of DILI**. (**A**) **Autoimmune-like DILI**. Reactive metabolites generated by CYPs and other drug-metabolizing enzymes can covalently bind liver proteins and form drug-modified self-antigens recognized by immune cells. Release of damage associated molecular patterns (DAMPs) resulting from hepatocyte stress induced by parental drug or their reactive metabolites can also contribute to the activation of immune cells. (**B**) **Steatosis**. Drugs may alter fatty acid (FFA) homeostasis in the liver by several mechanisms, including the reduction in FFA export as very low-density lipoproteins (VLDL), impairment of FFA entry in the mitochondria, direct inhibition of mitochondrial β-oxidation enzymes, disruption of mitochondrial respiratory chain (MRC) or depletion of mitochondrial DNA (mtDNA). Non-metabolized FFA are esterified to form triglycerides and accumulate in the cytoplasm of hepatocytes as lipid droplets. Drug-induced mitochondrial dysfunction can result in ATP depletion and increased formation of reactive oxygen species (ROS) and subsequent hepatocyte injury. Further activation of Kupffer and stellate cells by DAMPs and/or cytokines released by damaged hepatocytes may induce inflammatory and fibrotic responses in the liver and contribute to steatosis progression to steatohepatitis or fibrosis. (**C**) **Phospholipidosis**. Drugs can accumulate in lysosomes and decrease phospholipids (PLP) degradation by direct inhibition of phospholipases or by formation of degradation-resistant drug–PLP complexes. Drug-induced alterations of lysosomal function, with loss of hydrolytic enzymes due to release of enzymes or alterations in enzyme synthesis or transport, as well as changes in expression of genes involved in PLP metabolism, may result in PLP accumulation and formation of lamellar bodies. (**D**) **Cholestasis**. Drugs may disrupt homeostasis of bile acid in the liver by direct inhibition of hepatobiliary transporters located at the sinusoidal and canalicular membranes of hepatocytes or by alteration of key processes involved in the expression and/or localisation of transport proteins. Accumulation of bile acids up to cytotoxic concentrations can result in damage to parenchymal liver cells and bile ducts.

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
