# Peer review of "In Vitro Models for Studying Chronic Drug-Induced Liver Injury"

_ijms, 2022, doi:10.3390/ijms231911428_

Round 1

Reviewer 1 Report

The article entitled “In vitro models for studying chronic drug-induced liver injury” is an old issue in clinical hepatology. As more and more drugs are discovered or invented, the adverse effects of the drugs are manifested as drug-induced liver injury (DILI). Accordingly, several articles can be traced in literature about DILI. Also, there is no scarcity of articles about preclinical studies regarding models of DILI. Some of these are shown for ready reference (1. Kuna et al. Models of Drug Induced Liver Injury (DILI) - Current Issues and Future Perspectives. Curr Drug Metab. 2018;19(10):830-838. 2.Fernandez-Checa JC et al. Advanced preclinical models for evaluation of drug-induced liver injury – consensus statement by the European Drug-Induced Liver Injury Network [PRO-EURO-DILI-NET]. J Hepatol 2021;75:935-959. 3. Tasnim et al. .Recent Advances in Models of Immune-Mediated Drug-Induced Liver Injury. Front Toxicol. 2021 Apr 27;3:605392.).

Also, several others can be found. Additionally, cell0based in vitro models of DILI have been published ( Mirahmad et al. In vitro cell-based models of drug-induced hepatotoxicity screening: progress and limitation. Drug Metab Rev. 2022 May;54(2):161-193.).

In this perspective, it is important to assess the scientific impact of another article of the same or similar or related nature. In this context, the authors have focused on some factors that would be discussed in this article. They mentioned that this article will take care of “the major mechanisms implicated in DILI, highlighting the importance of chronic effects such as steatosis or cholestasis” the authors also opted to deeply discuss the emerging in vitro hepatic cell culture platforms and the technological methods for evaluating chronic DILI” (Final paragraph of INTRODUCTION). The authors of this article have published several articles about different aspects of DILI. They have comprehensively incorporated their experiences to compile the present article that is dedicated to “in vitro models of DILI”.

Specific comments

1.      The authors have discussed chronic drug-induced liver injury and Table 1 has shown the types. The mechanism underlying the induction of DILI by these drugs is not well understood. However, Discussion is required about the potential of drugs causing autoimmunity or hepatic steatosis or deposition, or cholestasis (Table1). The authors have tried to explain these mechanisms in Fig.1. My suggestion is to explore, for example how nitrofurantoin induces autoimmunity and such for some typical drugs or representative drugs.

2.      The entity, scope, and limitations of in vitro cell model should be discussed after the INTRODUCTION.

Author Response

We thank both reviewers for the constructive nature of their comments and suggestions on the manuscript, which have helped us to improve our resubmission.

Reviewer 1

  1. The authors have discussed chronic drug-induced liver injury and Table 1 has shown the types. The mechanism underlying the induction of DILI by these drugs is not well understood. However, Discussion is required about the potential of drugs causing autoimmunity or hepatic steatosis or deposition, or cholestasis (Table1). The authors have tried to explain these mechanisms in Fig.1. My suggestion is to explore, for example how nitrofurantoin induces autoimmunity and such for some typical drugs or representative drugs.

Following reviewer’s suggestion, we have including along the text examples of representative drugs causing steatosis, cholestasis and phospholipidosis through different mechanisms (Lines 221-228; ln 250-252, ln 265-6, ln273-279, ln283-292; ln315-321; ln327-329). In addition to this, in the case of autoimmune-like DILI we have explained how nitrofurantoin induces autoimmunity to exemplify this kind of chronic DILI (Lines 187-202).

  1. The entity, scope, and limitations of in vitro cell model should be discussed after the INTRODUCTION.

The scope, entity and limitations of in vitro cell models have been incorporated into the manuscript as suggested by the reviewer (Lines 410-414; ln 418-430). We have also introduced a new section (before the one illustrating the cell models) describing liver cell types and their contribution to DILI (lines 340-404).

Reviewer 2 Report

The review by Donato et al examines the various in vitro systems available to test chronic hepatotoxicity due to various insults. The review is generally well written, though the authors need to stress their focus on chronic liver injury throughout the review to differentiate it from other similar reviews already in the literature. Some additional concerns are also present.

Major comments

1) When discussing the in vitro systems, the authors should critically evaluate whether they replicate the in vivo pathophysiology, and if not clearly mention the drawbacks and suggest alternatives to work around these issues.

2) Though depicted in figure 1 to some extent, the authors neglect to detail the role of non parenchymal cells (NPC) in mediating chronic liver injury and that needs to be added, preferably as a separate section.

3) While mentioned in the section on co-cultures, the relevance of NPC co culture to the in vivo findings should be critically evaluated and discussed. Drawbacks of these systems, if any should be pointed out as well. For example, the authors mention the use of APAP in Insight spheroids (line 492), but neglect to point out that APAP induced injury would be an example of acute liver injury and not chronic injury which is the focus of the review. This caveat should be mentioned at every instance where APAP is used as model toxicant (eg. Line 524, 533). It would be helpful if the authors could thus focus on drugs which produce chronic injury which have been tested in such in vitro systems as well or else mention the lack of studies up front rather than at the end of the discussion (line 553).

4) Following up on this pattern, it would be useful to compare the discussed in vitro system with findings in vivo to let the reader know the relevance of these systems to the in vivo symptoms. For example, the authors mention the RegeneMed system which tested model compounds but make no mention of how this compares to the effects of these compounds in vivo.

5) The use of NPC in microfluidic systems and multi-organ chip systems seems to have been neglected and discussed. If studies are scarce, this caveat has to be clearly mentioned as well.

6) Sections 4.1 to 4.4 – It would be better if these sections could be incorporated into the sections discussing the relevant in vitro system, since the focus of the review is such systems relevant for study of chronic hepatotoxicity. As it stands, these sections are not very useful since it is not clear which of the in vitro systems previously discussed were used for the studies.

Minor comments

1) It might be useful to depict table 1 with columns and rows (4 columns with drugs listed in rows) rather than the current presentation which lists the drugs one after another.

2) Line 120- “….share features with many other…”

Author Response

We thank both reviewers for the constructive nature of their comments and suggestions on the manuscript, which have helped us to improve our resubmission.

Reviewer 2

The review is generally well written, though the authors need to stress their focus on chronic liver injury throughout the review to differentiate it from other similar reviews already in the literature. Some additional concerns are also present.

We agree with the reviewer that the although other authors have reviewed the existing cell models for hepatotoxicity studies, this review is focused on the evaluation of chronic hepatotoxicity and the existing in vitro models to study it, and we have better highlighted this introducing specific examples (i.e. nitrofurantoin) and specifying other important aspects.

Major comments

  1. When discussing the in vitro systems, the authors should critically evaluate whether they replicate the in vivo pathophysiology, and if not clearly mention the drawbacks and suggest alternatives to work around these issues.

As indicated by the reviewer, we have critically evaluated how the proposed models recapitulate the in vivo pathophysiology, including alternative to improve these test systems (Lines 418-430; ln444; ln 456-461; ln490-492; ln498; ln526-531; ln562-565; ln581-586; ln599-561; ln626-630; ln662-666; ln709-713).

2. Though depicted in figure 1 to some extent, the authors neglect to detail the role of non parenchymal cells (NPC) in mediating chronic liver injury and that needs to be added, preferably as a separate section.

We have included a new section about liver cell types, detailing the role of NPC in DILI as suggested by the reviewer (lines 340-404).

3. While mentioned in the section on co-cultures, the relevance of NPC co culture to the in vivo findings should be critically evaluated and discussed. Drawbacks of these systems, if any should be pointed out as well. For example, the authors mention the use of APAP in Insight spheroids (line 492), but neglect to point out that APAP induced injury would be an example of acute liver injury and not chronic injury which is the focus of the review. This caveat should be mentioned at every instance where APAP is used as model toxicant (eg. Line 524, 533). It would be helpful if the authors could thus focus on drugs which produce chronic injury which have been tested in such in vitro systems as well or else mention the lack of studies up front rather than at the end of the discussion (line 553).

Drawbacks of each in vitro system have been included in the text (Lines 418-430). In addition, we have included information about limitations of each model and have better highlighted the role of NPC (previous section + lines 714-719 + lines 840-842).

We agree that APAP is a model of acute toxicity. In order to avoid misunderstandings, we have included and highlighted the period of incubation. In most of the cases APAP is just used as a model example to prove that the cell model response to drug hepatotoxicants after acute or chronic exposure, and this has been clarified along the text (Lines 453-457; lines 696-700; lines 657-658).

4. Following up on this pattern, it would be useful to compare the discussed in vitro system with findings in vivo to let the reader know the relevance of these systems to the in vivo symptoms. For example, the authors mention the RegeneMed system which tested model compounds but make no mention of how this compares to the effects of these compounds in vivo.

Relevance of the models compared to the in vivo situation has been highlighted (ln444; ln 456-461; ln490-492; ln498; ln526-531; ln562-565; ln581-586; ln599-561; ln626-630; ln662-666; ln709-713) and in the case of RegeMed system we have also included the relevance compared to in vivo situation (i.e. Cmax) (Lines 662-670).

5. The use of NPC in microfluidic systems and multi-organ chip systems seems to have been neglected and discussed. If studies are scarce, this caveat has to be clearly mentioned as well.

We agree with the reviewer that NPC are scarcely described in the microfluidic systems. Although several approaches have used NPC, these models have not been used for the evaluation of chronic DILI, and thus, they are not detailed in Table 2. However, we have included this information on the text to avoid misunderstandings and we have included the need of NPC for better chronic DILI predictions (Lines 714-719).

6. Sections 4.1 to 4.4 – It would be better if these sections could be incorporated into the sections discussing the relevant in vitro system, since the focus of the review is such systems relevant for study of chronic hepatotoxicity. As it stands, these sections are not very useful since it is not clear which of the in vitro systems previously discussed were used for the studies.

In Table 3, we have included which relevant in vitro systems have been used with the different techniques proposed. Section 4 (now section 5) would like to exemplify which techniques are available for studying chronic toxicity. In general, most of the cell models used for hepatotoxicity assessments are based on the measurement of a single endpoint indicator of viability (MTT assay, ATP content, LDH release). It has been widely demonstrated that multiparametric assays better predict hepatotoxicity in vitro; thus, we would like to explain the techniques available although for some of them, just a single model has been used.

Minor comments

  • It might be useful to depict table 1 with columns and rows (4 columns with drugs listed in rows) rather than the current presentation which lists the drugs one after another.

We have changed Table 1 as indicated by the reviewer (page 3).

  • Line 120- “….share features with manyother…”

This sentence has been changed following reviewer’s indication (line 123).

Round 2

Reviewer 2 Report

The authors have responded to the queries satisfactorily